# First Conclusions on Damage Behaviour of Recycled Carbon Staple Fibre Yarn Using X-ray and Acoustic Emission Techniques

**DOI:** 10.3390/ma16134842

**Published:** 2023-07-05

**Authors:** Christian Becker, Joachim Hausmann, Janna Krummenacker, Nicole Motsch-Eichmann

**Affiliations:** Leibniz-Institut für Verbundwerkstoffe GmbH, 67663 Kaiserslautern, Germanynicole.motsch@ivw.uni-kl.de (N.M.-E.)

**Keywords:** recycled carbon fibre, rCF, composite material, in situ X-ray microscopy, acoustic emission, failure mechanism, recycling, sizing, fracture initiation and propagation

## Abstract

This paper presents the first results on the characterisation of the damage behaviour of recycled carbon fibre (rCF) rovings manufactured into unidirectionally (UD) reinforced plates. In the first step, the mechanical properties of several material combinations were determined by mechanical tests (tensile, flexural, compression). This proves the usability of the material for load-bearing structures. For example, a tensile modulus of up to 80 GPa and a tensile strength of 800 MPa were measured. Subsequently, the fracture surface was analysed by scanning electron microscopy (SEM) to characterise the fibre–matrix adhesion and to obtain first indications of possible failure mechanisms. Despite the high mechanical properties, poor fibre–matrix adhesion was found for all matrix systems. In situ X-ray microscopy tests were then performed on smaller specimens under predefined load levels as transverse tensile and bending tests. The results provide further predictions of the failure behaviour and can be compared to the previous test results. The three-dimensional scan reconstruction results were used to visualise the failure behaviour of the staple fibres in order to detect fibre pull-out and fibre or inter-fibre failure and to draw initial conclusions about the damage behaviour in comparison to conventional fibre composites. In particular, a benign failure behaviour in the transverse tensile test was demonstrated with this procedure. In addition, first concepts and tests for the integration of AE analysis into the in situ setup of the X-ray microscope are presented.

## 1. Introduction

At a time when energy efficiency and sustainability are of key importance, there is no alternative to fibre-reinforced plastic composites. In particular, carbon-fibre-reinforced plastics (CFRPs) play an important role in the automotive, aerospace, civil engineering and wind energy sectors [1,2,3,4,5,6,7]. However, as the demand for these materials increases, so does the amount of waste generated from end-of-life components and production waste. It is predicted that, in 2025, there will be 1700 t of CFRP waste from wind energy alone, rising to 5300 t in 2035. Looking at the CO_2_ emissions from the production of virgin fibres alone, 45.26 kg is released when 1 kg of carbon fibre is produced. In comparison, only 6.91 kg is released in a pyrolysis process to obtain recycled carbon fibres (2.70 kg in solvolysis). By recycling carbon-fibre-reinforced plastics, the fibres can be obtained in a more environmentally friendly way than by producing new fibres and the material cycle can be closed. The use of recycled fibres can therefore reduce the production of new fibres to the extent that they can be replaced [8,9,10,11,12,13].

The recycling of CFRP is challenging both in terms of isolating the fibres from the matrix and in terms of reusing the reclaimed fibres in a technical or structural application with load-bearing requirements. Various pyrolysis and solvolysis processes offer the possibility of separating the fibres from the matrix. When considering recycling methods for carbon-fibre-reinforced plastics, the following materials can be recycled [14,15,16,17,18,19].

-Pre-impregnated fibres (i.e., prepreg offcuts).-Cured fibre composites (e.g., end-of-life components, production scrap).

Depending on the material present, a suitable process must be selected and the process parameters must be adjusted to recycle the fibre composite and obtain recycled carbon fibres. Composites with thermoplastic or cured thermoset matrices, as well as pre-impregnated fibres, can be freed from the matrix material by means of pyrolysis, electrodynamic fragmentation or solvolysis, among others. In all cases, the technical requirements of the process lead to the shredding and thus shortening of the fibres [14,20].

Subsequent continuous fibre reinforcement, as with virgin fibre components, is not possible due to the shortened fibres. The shortened rCF fibres have to be converted into other types of semi-finished products, such as roving into a continuous staple fibre yarn [21]. One product that is commercially available is an rCF roving from the Wagenfelder Spinnerei GmbH (Germany) [22]. Other products, not discussed in this paper, are woven or non-woven fabrics or their use in SMC products [17,23].

Knowledge of the mechanical and damage behaviour is important for the design of components. The characterisation of mechanical properties by means of standardised tests (tensile, compression, etc.) is well established [21]. More complex tests have to be performed if the damage behaviour is to be analysed. In situ X-ray experiments are an excellent method for performing mechanical tests and detecting structural changes in the meantime. Especially for rCF fibres, they can be an important source of information, e.g., on the exact fibre orientation, the crack initiation and propagation, the fibre pull-out and post-failure behaviour [24,25,26,27,28,29,30]. Another advantage is that the 3D CT volume scans can be subsequently integrated as a model into simulation software and can thus be used again as a basis for the design of components [29].

The motivation for the work presented in this paper is, on the one hand, the characterisation of the mechanical properties of the rCF staple fibre yarn with different matrix systems and the associated assessment of the usability in load-bearing structures. Furthermore, the influence of the sizing on the mechanical performance will be investigated. To what extent does the presence or absence of sizing affect the properties and how does it affect the damage behaviour? For this purpose, extensive characterisation of both the dry and embedded fibre was carried out before and after the mechanical tests using Fourier transform infrared spectroscopy (FTIR), micrographs and scanning electron microscopy (SEM). Finally, in situ X-ray microscopy experiments were carried out to evaluate the failure behaviour and the associated influence of the sizing.

The questions that arise here are what kind of failure event will occur? Is it more likely to be fibre-to-fibre or inter-fibre breakage? Is the resulting failure behaviour comparable to that of a UD composite or is it more similar to the behaviour of short fibre-reinforced composites? Does fibre slippage occur under load? Does fibre pull-out occur and is there sufficient fibre–matrix adhesion? Does fibre alignment occur under load? More generally, we want to find out what damage mechanisms can be detected by in situ X-ray microscopy.

## 2. Materials and Methods

### 2.1. Description of rCF Material and Processing of Samples

The staple fibre yarn used from Wagenfelder Spinnereien GmbH (Wagenfeld, Germany) (Figure 1) consists of 60–80 mm long individual fibres that are twisted in a carding and spinning process (similar to rope production) and then fixed by a polyamide 6 (PA6) thread (binding yarn) [15,22]. In addition, PA6 filaments of the same size as the rCF fibres are inserted. This allows for a variation in the rCF/PA6 ratio. In this work, a 90/10 (rCF/PA6) yarn was used. In the following, this rCF yarn is referred to as roving. According to the manufacturer, the rCF fibres used come partly from production waste, but also from end-of-life components that have been shredded and pyrolysed. The peculiarity of the roving due to the manufacturing process is that it does not have a straight alignment but an undulated shape (Figure 1). This undulation has a theoretical influence on the mechanical properties in the processed state, as deviations in the fibre orientation lead to a decrease in the mechanical properties. To counteract this effect, attempts were made to pre-tension the rovings during the manufacturing process (filament winding) to compensate for the undulation and to align the fibres. Micrographs and fibre orientation analyses using X-ray CT scans are shown in Chapter 3. To achieve the best possible alignment during the filament winding process, tensile tests were first carried out on the dry, unimpregnated rovings. This was performed using a tape tensile tester developed at the IVW that, by deflecting the tapes for the purpose of restraint, reduces excess stress in the restraint and determines the pure tensile strength of the tape [31]. This effect can also be used for the rovings.

The test setup and the determined force–elongation curves are shown in Figure 2.

Due to the inhomogeneity of the rovings, the resulting tensile strength (in N) is widely scattered between 10 and 18 N. Therefore, the lower limit of 10 N was used as the pre-tensioning force for the subsequent panel production to avoid possible cracking or slipping of the rovings.

Further analysis of the rovings was carried out using Fourier transform infrared spectroscopy (FTIR) and scanning electron microscopy (SEM) on the dry rCF fibres to characterise any residual sizing, matrix or foreign matter that may reduce the fibre–matrix adhesion. For FTIR, an amount of 5 g of fibres was placed in acetone for 24 h to remove contaminants from the rCF rovings. After evaporation of the acetone, the remaining material was analysed. The results of this test and the corresponding SEM images are shown in Figure 3. The FTIR analysis showed a 70–80% match with an epoxy sizing. This is a first indication that the fibres used are either not pyrolysed or have been re-coated with a new sizing. Unfortunately, there is no reliable information from the manufacturer.

A film can be clearly seen on the fibre when looking at the SEM image (Figure 3(b1)). This again indicates that there is no untreated fibre. In addition to the film, damage is also visible on the fibre, possibly caused by the carding process (Figure 3(b2)). Such pre-damage may have a negative effect on the subsequent mechanical properties, but has not been considered further below.

Subsequently, the samples were produced using a plate winding process. There are a number of different production methods, depending on the resin system (described below). Three different resin systems were used. The aim was to characterise the influence of the sizing of the rCF fibres (possible influence by the pyrolysis process) on the mechanical properties and the failure mechanism. The resin systems used with the corresponding manufacturing process in the winding process and their sample declaration are shown in Table 1.

The first step is to examine the microstructure of the plates produced by micro-sectioning. The impregnation quality, possible pore content and fibre volume content are to be determined approximately. The impregnation of the EP-rCF, PA6-rCF and the Bio-rCF specimens is of good quality (cf. Figure 4). There are no large defects or pore accumulations that could affect the mechanical properties. The fibre volume content differs slightly between the two samples (Bio-rCF ≈ 37% and EP-rCF and PA6-rCF ≈ 30%), which is due to the different processing methods. This must be taken into account when interpreting the results.

### 2.2. Description of Mechanical Tests

In order to obtain an idea of the failure behaviour of the unidirectional rCF plates, static mechanical tests were carried out. In addition to the characteristic values obtained, the fracture surfaces were analysed and compared using SEM images. This allows fibre fracture, matrix failure or fibre pull-out to be distinguished. In addition, the mechanical properties can be used to draw initial conclusions about the fibre–matrix bonding and the compatibility of the sizing with the resin system.

The tests on the samples of all resin systems were:-Tensile tests according to EN ISO 527-5:1997 [32];-Compression tests according to EN ISO 14126:1999 [33];-Bending tests according to EN 2562:1997 [34].

All measurements were tracked with an optical measurement system to determine strain more accurately.

For the time being, the focus is on the tensile and flexural properties of the material, as a unidirectional structure is being tested. The compressive properties were also tested for completeness, but will not be discussed in detail later as no in situ compression tests have yet been carried out.

### 2.3. Description of X-ray Microscopy and Acoustic Emission

In order to detect and characterise the damage and failure behaviour in addition to the mechanical parameters, in situ X-ray microscopy tests were carried out on the Bio-rCF and the EP-rCF. For this purpose, the test setup, the test procedure and, above all, a suitable specimen geometry have to be determined. No standard can be used for this series of tests.

The test set-up for the in situ tests is shown in Figure 5. A material testing device from Deben UK Ltd. (Bury Saint Edmunds, UK), specifically designed for in situ computed tomography (CT) testing, was used [35]. It can be used for static and low-cycle tensile, compression and bending tests up to a maximum load of 5 kN. This device was built into the ZEISS Xradia (Oberkochen, Germany) 520 Versa X-ray microscope [36]. Special optics inserted in the beam path allow resolutions of up to 200–300 nm to be achieved. In the in situ experiments carried out in this work, measurements were made with a resolution of 1.5 μm.

The first step was to design a specimen geometry that would not exceed the maximum load to failure, but would also achieve the required resolution while still containing sufficient material structure (at least one complete roving). Different geometries were milled from a wound rCF plate and tested (cf. Figure 6). The sample geometry in Figure 6b showed the best conditions due to its resolution and failure behaviour. However, the maximum load in the fibre orientation (0° orientation) was challenging for all specimen geometries. Unfortunately, the 0° tests resulted in increased failure in the clamping as well as slipping in the clamping (clamping force could not be increased due to the associated damage to the specimen). Therefore, at this stage, only tensile tests transverse to the fibre orientation could be carried out. In the discussion in Chapter 4, the first approaches to solving the problem are listed, which can also be solved by the installation of the AE measurement technology.

At first, the tensile properties of the in situ samples were measured ex situ with the in situ test device to determine the load levels for the actual measurements. This is important for finding relevant points, where the actual CT scan can be started to detect the failure mechanism. It is also important to find the relaxation time of the specimen at certain load levels in order to reduce specimen movement during the CT scan [37]. The results for both materials are shown in Figure 7a and the diagram of the in situ measurement with the corresponding load steps is shown in Figure 7b. The relaxation time for both materials was approximately 30 min after reaching the target load. No further changes in load were measured after this time. Relaxation in the Bio-rCF tests was different at all load levels. This could be due to the small number of load steps and the therefore greater differences in applied loads from one step to another. In the EP-rCF tests, the relaxation was around 10–15% of the target load (except for load step 2 at 30%).

From these tests, the stress levels were selected and the relaxation range and time at these levels were determined:

The same procedure was followed for the bending specimens (Figure 8). The specimen geometry was a rectangular specimen with dimensions of 25 × 3 × 2 mm (length × width × height) and a support width of 20 mm. Seven load levels were set to give a better overview of fibre movement, likely delamination or crack initiation.

## 3. Results

The results of the quasi-static mechanical tests, the associated microscopic examination of the specimens and fracture surfaces and the in situ X-ray microscopy tests are presented below.

### 3.1. Mechanical Behaviour under Static Load and Microscopic Analysis

As mentioned in Section 2.2, the mechanical tests were carried out in accordance with the standards EN ISO 527-5 (tension), EN ISO 14126 (compression) and EN 2562 (bending). All tests were recorded with an optical measuring device and the strain was evaluated by digital image correlation.

For all samples, tensile tests were carried out. Due to the unavailability of the matrix material, compression and flexure tests could only be carried out on the EP-rCF and PA6-rCF samples. However, the tensile properties are the most relevant and informative for the initial assessments and the following in situ tests. The influence of the matrix system on the properties can nevertheless be determined and assessed through the mechanical tests and fracture surface analysis.

#### 3.1.1. Tensile Properties

Looking at Table 2, the properties of EP-rCF and PA6-rCF are close to each other. The strain at break is almost the same. Young’s modulus and strength are approx. 16% apart. This may be due to the slightly different mechanical properties of the matrix systems. The Bio-rCF values are significantly higher (Young’s modulus ≈ 80,000 MPa and strength ≈ 850 MPa) due to the higher fibre volume content, but show that an increase of approximately 10% in fibre volume content leads to an increase in stiffness of up to 70% (for PA6-rCF) and an increase in strength of up to 76%. The SEM images of the failed samples (Figure 9) show poor fibre–matrix bonding for all matrix systems. Looking at the SEM image of the dry rCF fibre and the FTIR analysis (cf. Figure 3), it might be expected that the bonding of the rCF fibres to the EP matrix would be better than to other matrices, but this is not the case.

For the Bio-rCF and the EP-rCF samples, large fibre channels can be seen, which are due to pure fibre pull-out. In addition, no matrix attachment is visible on the exposed fibres. Gaps between the fibre and matrix are also visible on the visible embedded fibres. This could be due to delamination by the applied stresses. However, it is also possible that the matrix adhesion was not well established during the manufacturing process. In contrast, the fibre–matrix adhesion in the PA6-rCF is slightly better. Fibre channels are also visible, but more fibres are embedded and fibre–matrix adhesion is visible on the exposed fibres. The fibre–matrix adhesion is not ideal, but looks better than with the other system, which is in contrast to the FTIR results.

#### 3.1.2. Bending Properties

In the following, the flexural properties of the EP-rCF and PA6-rCF specimens are presented and their characteristic values are compared. The stress–strain diagrams (Figure 10) show an apparently linear course up to a strain of approx. 1.2%. After that, a decreasing slope of the graphs up to fracture is visible. This could be a consequence of the matrix influence of the composite, which is favoured by the mobility of the staple fibres in the material.

Table 3 lists the mechanical flexural properties compared against each other. It can be seen again that the composite with the epoxy resin matrix has the better mechanical properties (approximately 14% higher flexural stiffness and approximately 17% higher flexural strength). The elongation at break is approximately 1.5% for both.

When looking at the SEM images (Figure 10) of the flexural specimens, similar characteristics to those present in the tensile specimens are noticeable. Fibre pull-out channels are evident in both materials, indicating poor fibre–matrix adhesion. The exposed fibre ends of the EP-rCF specimens do not exhibit matrix adhesion, again confirming this. In the PA6-rCF samples, however, isolated adhesions are visible, again indicating slightly better adhesion properties to the rCF fibre.

#### 3.1.3. Compression Properties

In accordance with the standard, the specimens fail by wedge-shaped splitting or by continuous thickness shear. The mechanical properties (Table 4) of the PA6-rCF, with a 37,303 MPa stiffness and 325 MPa compressive strength, are approximately 10 and 13%, respectively, below those of the EP-rCF (approximately 41,200 MPa (stiffness) and 375 MPa (compressive strength)).

### 3.2. In Situ Failure Characterisation

In the following, the results of the in situ X-ray microscopy tests are presented. First, the transverse tensile tests of the bio-resin and the epoxy resin system are shown. Subsequently, a bending test transverse to the fibre orientation of the bio-resin system is explained as an example. The respective load levels are shown in Chapter 2 and will not be repeated here.

#### 3.2.1. Transverse Tensile Behaviour

Beginning with the results of the testing of the Bio-rCF, the following behaviour, which can be seen in Figure 11, was detected.

In the first load step (100 N), a small crack (30 μm) appears at the edge of the specimen. This crack spreads through the matrix with a higher load (crack length of 124 μm at 170 N). It can be seen that the crack propagates between the roving structures. In addition, bigger resin pockets were not affected by crack initiation or propagation. At the third load level (220 N), a nearly horizontal crack through the whole specimen at the smallest diameter of the sample geometry can be observed. A reason for crack propagation from load level one to three is that the crack propagation is independent of the material structure (parts of higher/lower fibre content) and the tailored shape of the specimen promotes a defined failure event at the middle of the sample. The sample geometry is therefore suitable for receiving a defined failure event at the centre plane and can be reliably detected in the region of interest in the X-ray microscope. 

At the fourth load level, a crack opening of 0.1 mm was performed to reconstruct the fibre pull-out while post-failure. Here, it can clearly be seen that isolated fibres cross the crack and connect the two separated parts. A fibre pull-out because of free fibre ends in the crack zone cannot clearly be detected because of the small elongation of the sample (overlapping length of the fibres up to 50 mm). This will always lead to a compromise: by increasing the field of view to detect a larger pull-out length, the resolution will downgrade and single fibres cannot be detected. Only an estimation can be made with the in situ tests and it has to be proved in standard tensile tests.

The same behaviour can be detected in the EP-rCF sample. In this test series, eight load steps were performed. This resulted from the difference in applied load on the sample in the in situ measurement and the pre-tests. Relaxation processes during the CT measurement (load drop can be seen in Figure 7 bottom) and the resulting stress relief in the material are causal for this increase in load steps because failure occurs at a later state. The increase of 40% from the mean value of pre-tests (16% from max. value) can be seen in Figure 7b. For comparison purposes, only the last 4 load levels are shown in Figure 12, which illustrates the failure behaviour for the EP-rCF.

Here too, the damage behaviour is similar to that of the Bio-rCF. In Figure 12a, a slight crack is visible (marked in red for better visualisation). The crack initiation starts within a roving that is slightly above the narrowest part of the specimen (normally, the maximum stress under tensile load is in the narrowest part of the tapered specimen). When the load is increased to 300 N, the crack continues to expand through the roving (Figure 12b). In Figure 12c at 350 N, failure of the specimen takes place. It can be seen that the crack that penetrates the entire specimen is not identical to the crack seen at the previous load levels. This suggests that micro-cracks are already present in the previous load levels that were not visible in the CT scan but lead to failure due to the stress maximum at the narrowest point. Figure 12d shows post-failure, where the crack has widened by 0.1 mm. It can be seen that individual filaments bridge the crack and thus maintain the structural cohesion of the specimen (“fibre bridging”). No more load is applied to the specimen (constant load at 20 N) and only fibre pull-out from the interconnected fracture surfaces takes place. This behaviour can also be verified by SEM images of the failed specimens from the quasi-static mechanical tests, where fibre bridging also occurred (cf. fibre channels visible)

#### 3.2.2. Transverse Bending Behaviour

As one example, the transversal bending behaviour of the Bio-rCF material was tested in situ to visualise the behaviour of the rCF. Due to the fact that the tensile behaviour of both materials is similar, only one sample was tested because of the long measurement time.

Figure 13a shows that a crack initially forms on the upper side of the sample near the pressure fin, starting from the surface. Since the crack runs diagonally through the material, the crack can only be shown in a 2D sectional image for this paper (image resolution in which the connection to the surface is not visible). In Figure 13b, this connection is now visible due to the crack growth. At both load levels, pure compression failure near the compression fin can be concluded. Only at load level 4 (Figure 13c) can the first cracks be detected on the tension side of the specimen, which are due to tensile stresses. At load level 6 (Figure 13d), it can clearly be seen that the cracks from both the top and bottom of the specimen propagate in a straight line through the material, leading to the failure of the specimens. Delamination and shear failure are not visible.

### 3.3. Inclusion of AE in In Situ X-ray Microscopy Inspection

In order to integrate the sensors of the acoustic emission (AE) apparatus into the Deben CT 5000 in situ module, new clamps must first be developed that allow the sensors to be guided through the closed construction into the interior. The clamps were designed in such a way that a new sample geometry can be used [38]. The setup including the specimen can be seen in Figure 14.

First AE tests (no in situ tests) on the rCF test specimens showed that the general setup works. The sensor positioning is sufficient for obtaining a large scan field of the sample. In addition, the sensor position delivers conclusive signals. The results of one Bio-rCF sample with the new test setup and sample geometry are shown in Figure 15. In order to check the suitability of the specimen geometry for UD tests in 0° orientation, no transverse tensile tests were carried out in contrast to the tests carried out above. The fibre orientation was aligned in the direction of the tensile stress.

The typical stress–strain curve for the rCF material (blue line) is shown. The orange dots represent the recorded AE events, which provide clues to events in the material. Here, it can be seen that, up to an elongation of 0.13%, no conspicuous signals were recorded. From there on, not only does the number of recorded events increase (increase in point density), but there is also a clear increase in the amplitude of the signals, especially from 0.15% strain. This increase correlates with the onset of damage and ultimately the failure of the specimen at 0.17% strain in the tensile test. This in turn means that the acoustic signals have detected initial damage in the material.

In order to obtain more precise conclusions and interpretations about the AE signals of the rCF samples, more measurement series are planned for the future. These should show statistically representative statements about possible damage events that are characteristic for the rCF material. When these data are available, first, coupled in situ measurements can be carried out. By describing the AE signals for the rCF material more precisely, it is then possible to determine the load levels more precisely in the in situ tests and to record the crack initiation and crack growth on the basis of the signals determined during the tests. These results can then be taken as a basis for automating the test procedure to reduce time and also to reduce time-dependent effects on the material during the test series (creep, relaxation).

## 4. Discussion

The mechanical behaviour of the unidirectional rCF material from rCF rovings shows some remarkable properties. The tensile modulus is close to those of composites with virgin fibres (Table 5), despite the poor fibre–matrix adhesion demonstrated in microscopic tests.

Failure in bending tests shows a combination of fibre and inter-fibre fractures. Mainly compression (upper side of the sample) and tensile failure (lower side of the sample) with subsequent fibre pull-out occur. Delaminations are not visible. The compression failure generally corresponds to those of a classic continuous fibre-reinforced composite. The variations that occur within but also in comparison between the matrix systems can be related to the inhomogeneity of the rovings, which leads to a different distribution and orientation of the fibres in the component. However, the slightly different processing methods can also influence this.

The in situ tests shown above give a first insight into the static failure behaviour of rCF staple fibre yarn depending on the fibre orientation. Comparing this behaviour with conventional unidirectional CFRP components, some differences can already be seen. In CFRP components, mainly matrix failure occurs in transverse tension, and the fibres remain unaffected. This means that the applied load is taken up by the matrix and that crack formation and propagation take place exclusively within the matrix and the fibre–matrix interface. In the rCF specimens, crack initiation and propagation also occur within the matrix. However, due to the inhomogeneity of the staple fibres (protruding fibre bundles, undulation of the yarn), a larger number of fibres are aligned in the direction of the load and thus also absorb some load. This leads to higher transverse tensile strengths, but also to a different post-failure behaviour of the material.

The failure that occurs can be described as good-natured, as individual fibres or rovings bridge the fracture and thus there is no complete structural failure (Figure 16a,b). The post-failure behaviour is characterised by a forceless fibre pull-out of the bridging fibres (due to the lack of fibre–matrix adhesion). This means that, depending on the length of the bridging fibres, the elongation of the specimen after failure can be up to 50% of the individual fibre length (here, between 30 and 40 mm) until structural failure occurs.

The effect of good-natured failure was also demonstrated in the in situ bending tests (Figure 16c). In addition, the failure behaviour (failure under compression and tensile load as a function of the fin arrangement) from the classic bending tests could be verified.

## 5. Conclusions

Several series of tests were carried out to determine the failure behaviour of rCF staple fibre yarn. After the characterisation of the rCF staple fibre yarn and the sizing, panels with three different matrix systems were produced in a winding process and then extensively characterised. Mechanical tests (tensile, compressive, bending) were carried out to determine the general material properties and to obtain first impressions of the failure behaviour from the fracture surface using SEM. In situ X-ray microscopy tests were then carried out to characterise the failure behaviour in detail. More precisely, transverse tensile tests and bending tests were performed.

The results described above on the mechanical behaviour of rCF staple fibre yarn un-der quasi-static loading provide a first approach to describing the failure behaviour of this material. The mechanical properties, particularly the high Young’s modulus, indicate the potential of the material for use in load-bearing structures. In addition, the benign failure behaviour transverse to the fibre orientation, which has been determined so far, offers advantages in load-bearing structures for avoiding abrupt failure and thus minimising the risk or reducing the safety factor in the design of the components.

Despite the positive aspects, the main drawbacks are the lack of information on the origin and history of the rCF fibres used in the yarn and, consequently, the poor fibre–matrix adhesion due to missing or inappropriate fibre sizing. This would require the incorporation of production steps, both in the manufacturing process and in the further processing of the rCF staple fibre yarn, which would lead to improved adhesion. This could further improve the mechanical properties. However, it is questionable whether this would lead to a loss of the “good-natured” failure behaviour and whether the fibre pull-out effect would be replaced by a higher probability of fibre breakage.

On the experimental side, the combination of AE and XCT can increase the information density and accuracy on the failure behaviour. The combination can also significantly reduce the measurement time for a series of tests, as the CT scans are started when a failure event occurs in the material, eliminating the need for preliminary tests or a careful approach to the fracture strength of the specimen being carried out. As described above, initial tests are already underway, including automation of the test procedure.

The specimens used in Chapter 4 in in situ tensile tests result in (a) a defined gauge length in which the specimens fail, (b) the specimen geometry is also suitable for testing in the 0° direction (no failure in the restraint) and (c) the specimen can be used both in situ and ex situ, thus providing a greater comparability of results and facilitating preliminary testing.

The approach of combining AE and in situ CT does not yet provide meaningful results. However, it has been shown that the approach and the sample geometry used are suitable. Further investigations with the rCF material may provide a deeper insight into the failure behaviour and whether and to what extent pure matrix failure with fibre pull-out or additional fibre fractures occurs.

## Figures and Tables

**Figure 1 materials-16-04842-f001:**
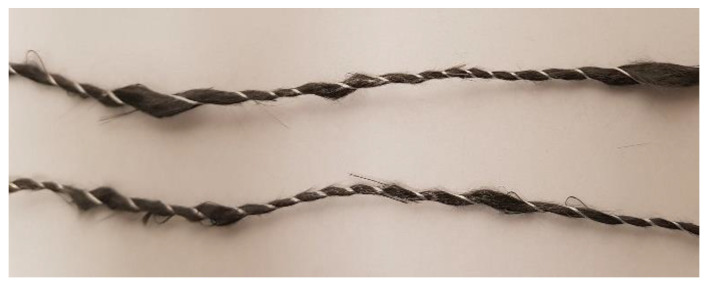
rCF roving with binding yarn leading to undulation and inhomogenisation of fibre thickness; a few protruding fibres can be seen.

**Figure 2 materials-16-04842-f002:**
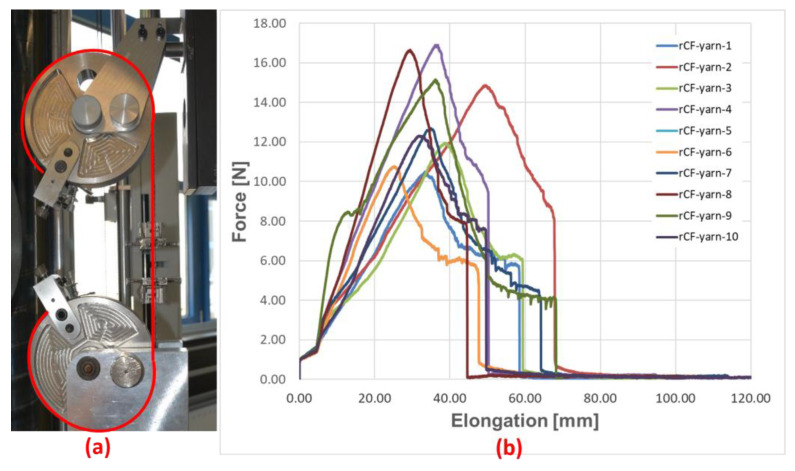
Dry tensile tests on rCF rovings: test setup with tape clamping system and clamping of roving (red line) (**a**); force–displacement diagram (**b**).

**Figure 3 materials-16-04842-f003:**
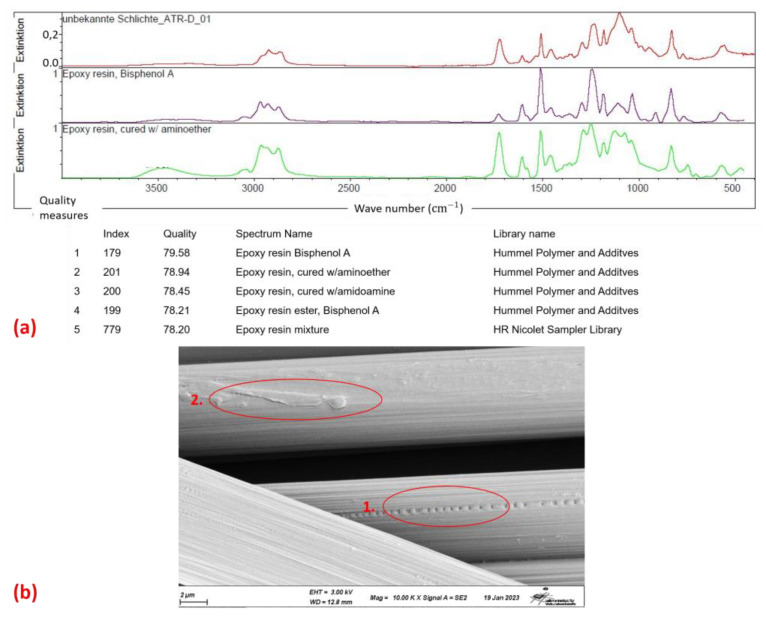
Analysis of dry rCF roving: FTIR analysis of residues on fibres (**a**) and SEM images of fibre surface with film attachment (1) and damages on fibre (2) (**b**).

**Figure 4 materials-16-04842-f004:**
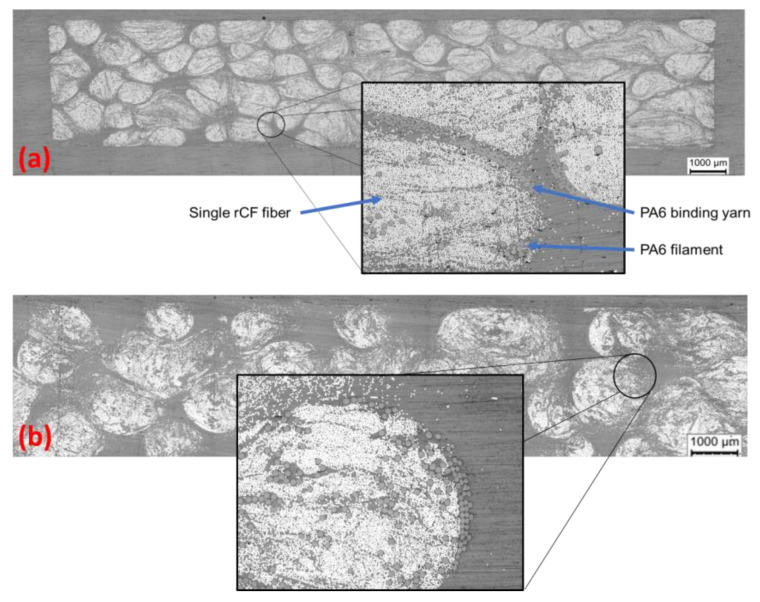
Microsection of the Bio-rCF (**a**) and EP-rCF (**b**). Typical microstructure with matrix, single rCF fibres, binding yarn and PA6 filaments visible.

**Figure 5 materials-16-04842-f005:**
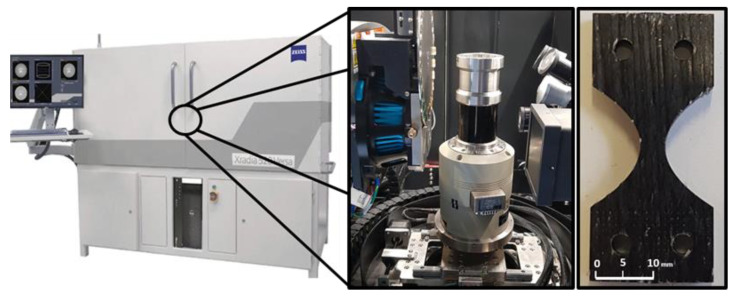
X-ray Microscope Zeiss Xradia 520 Versa with in situ Module Deben CT 5000 and used sample geometry.

**Figure 6 materials-16-04842-f006:**
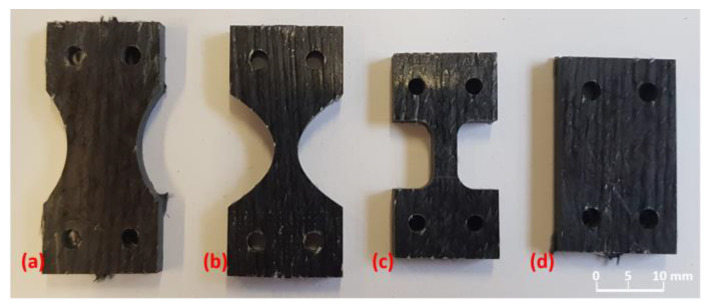
Different sample geometries tested in situ: (**a**) tapered sample with large region of interest (ROI), (**b**) tapered sample with small ROI, (**c**) nearly straight ROI sample and (**d**) rectangular sample.

**Figure 7 materials-16-04842-f007:**
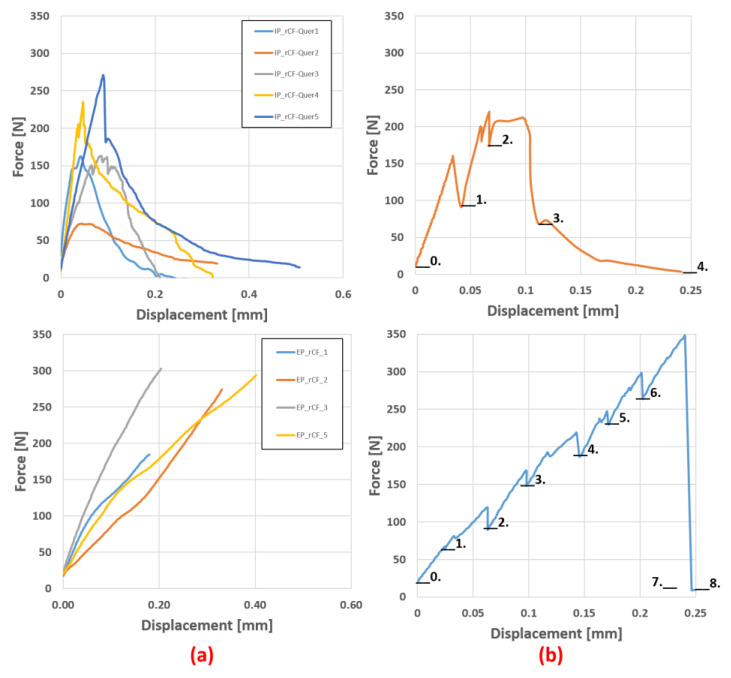
(**a**) Force–displacement diagram from the pre-tests of the rCF in situ samples Bio-rCF (top) and EP-rCF (bottom) normal to the fibre orientation. (**b**) Measurement diagram during in situ experiments with load levels; relaxation of sample visible.

**Figure 8 materials-16-04842-f008:**
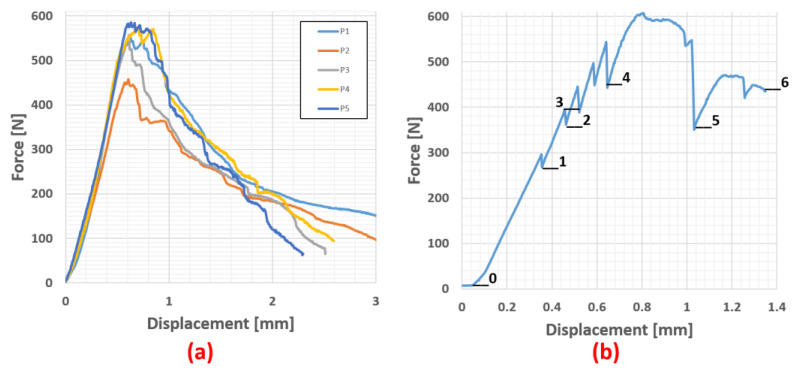
(**a**) Force–displacment diagram of the Bio-rCF bending pre-test and (**b**) measurement diagram during in situ bending experiment.

**Figure 9 materials-16-04842-f009:**
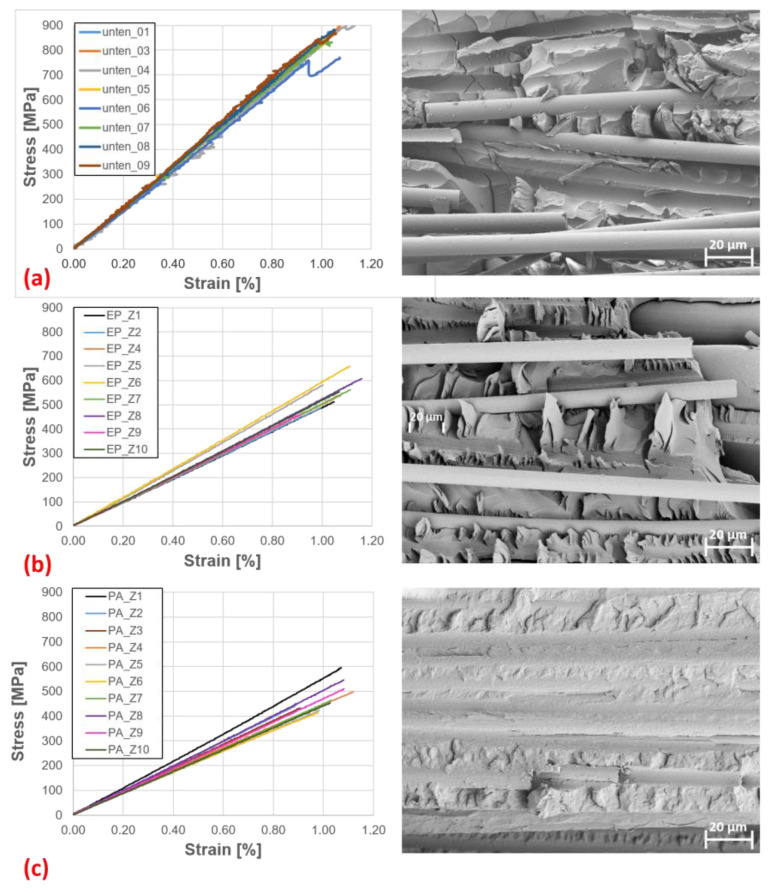
Tensile tests: Stress-strain diagrams and SEM images of fracture surfaces of (**a**) Bio-rCF, (**b**) EP-rCF and (**c**) PA6-rCF.

**Figure 10 materials-16-04842-f010:**
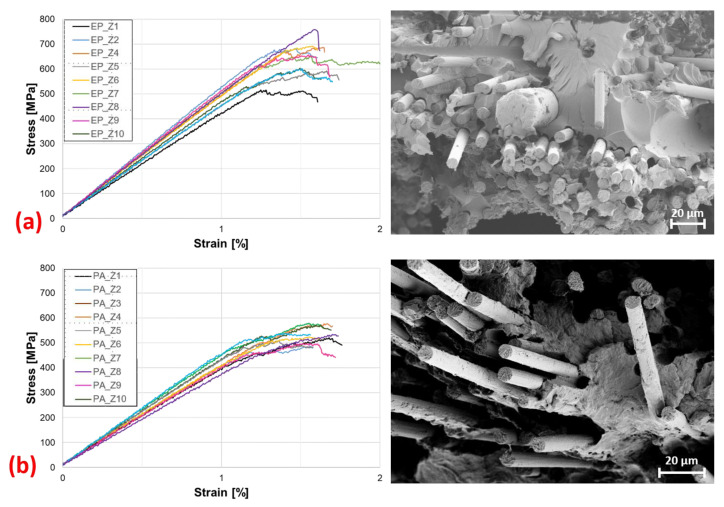
Bending test: stress–strain diagrams and SEM images of fracture surfaces of (**a**) EP-rCF and (**b**) PA6-rCF.

**Figure 11 materials-16-04842-f011:**
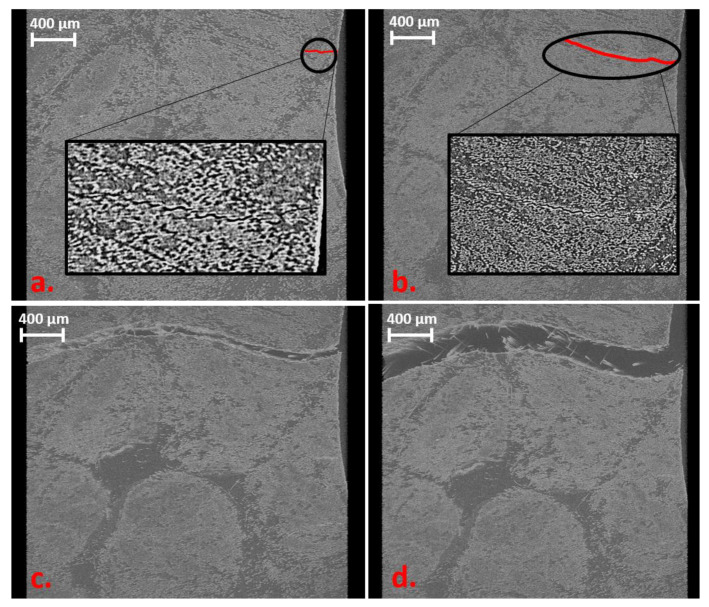
Volume rendering images of the different load levels (unloaded level not shown) of Bio-rCF sample. (**a**) 100 N: small crack (red line) at the edge. (**b**) 170 N: crack propagation (red line) between the fibres. (**c**) 220 N: failure of the sample. (**d**) Crack opening of 0.2 mm: some bridging fibres visible.

**Figure 12 materials-16-04842-f012:**
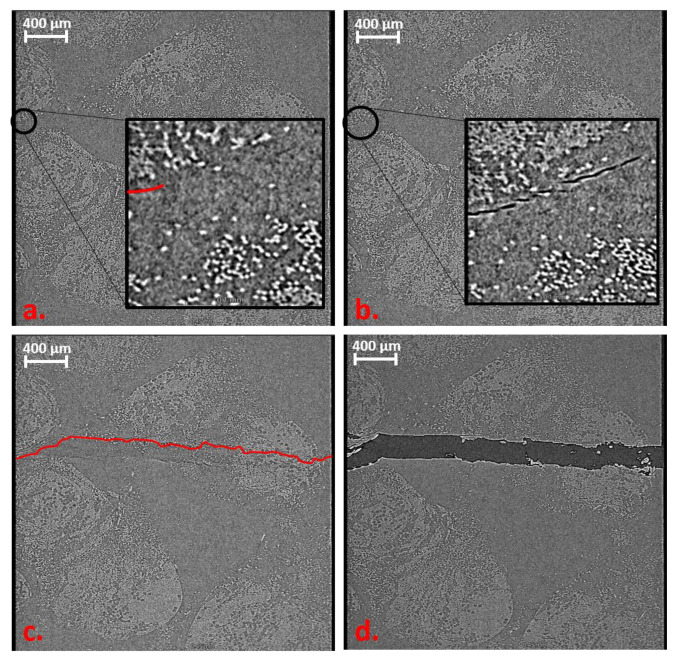
Volume rendering images of in situ tensile test of EP-rCF. (**a**) 220 N: small crack (red line) at the edge of sample. (**b**) 250 N: crack propagation between the fibres. (**c**) 300 N: failure of the sample. (**d**) Crack opening of 0.2 mm.

**Figure 13 materials-16-04842-f013:**
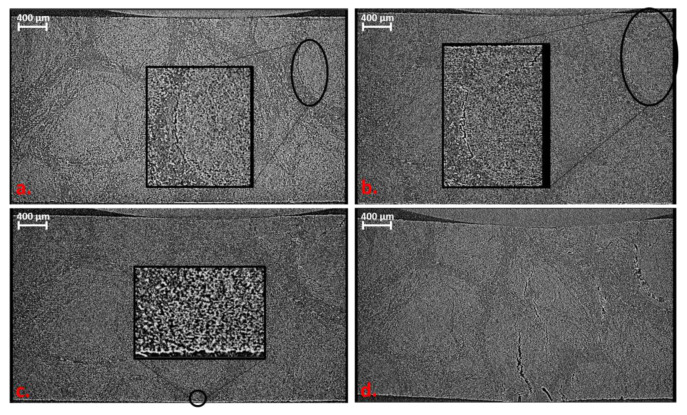
Volume rendering images of in situ bending test of Bio-rCF. (**a**) 400 N: small crack at border to the pressure fin. (**b**) 500 N: crack propagation between the fibres. (**c**) 550 N: crack initiation on tension side of sample. (**d**) 600 N: failure of sample before load drop.

**Figure 14 materials-16-04842-f014:**
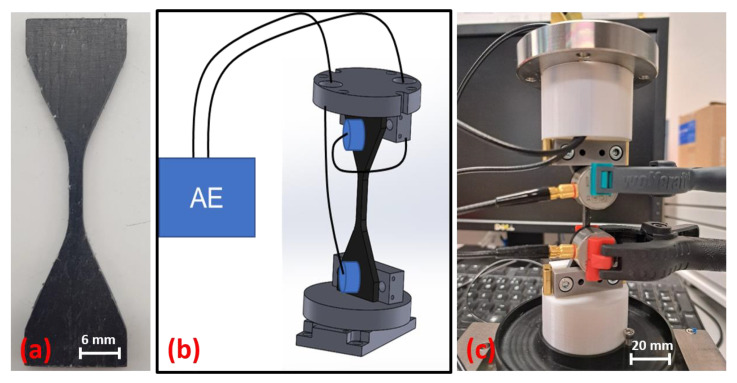
Integration of AE sensors into in situ Deben CT 5000: (**a**) adapted sample geometry, (**b**) construction drawing of setup and (**c**) first integration try with in situ test device.

**Figure 15 materials-16-04842-f015:**
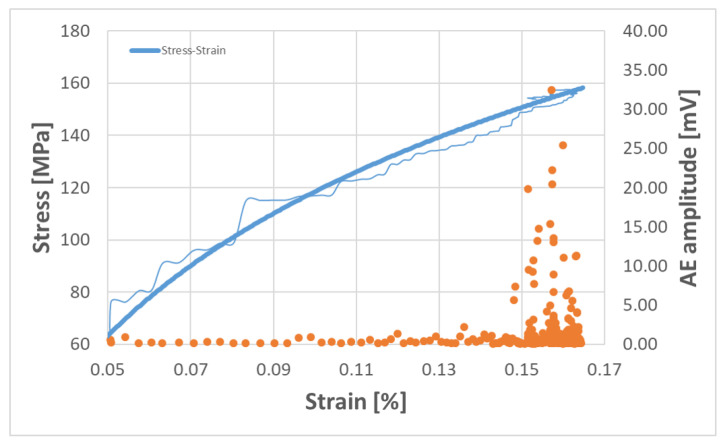
Example of stress–strain diagram (blue lines; bright line average value) of Bio-rCF sample with AE signal distribution (orange dots).

**Figure 16 materials-16-04842-f016:**
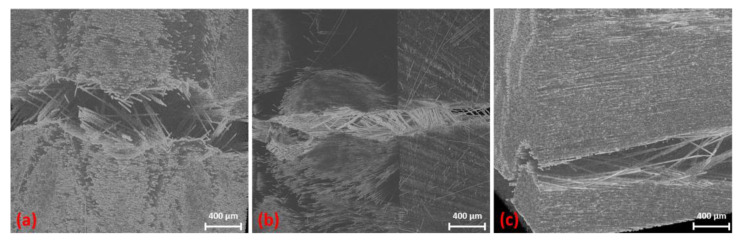
Good-natured post-failure behaviour with bridging fibres detected in in situ tests: Bio-rCF tensile test (**a**), EP-rCF tensile test (**b**) and Bio-rCF bending test (**c**).

**Table 1 materials-16-04842-t001:** Different material combinations of rCF and resin used in this paper compared with their manufacturing process.

Name	Resin System	Producer Name	Manufacturing Process
EP-rCF	Thermoset	Huntsman (The Woodlands, TX, USA) Araldite^®^ LY 1135-1; hardener Araldur^®^ 917; accelerator 960-1; mass ratio 100:90:2	wet winding
Bio-rCF	Bio Thermoset	UP resin ENVIREZ 70301 from Ashland (Wilmington, DE, USA); 22 MA-% bio content	dry winding process with subsequent tempered manual impregnation
PA6-rCF	Thermoplastic	PA6 film produced at IVW (Ludwigshafen, Germany) from Ultramid^®^ B36 LN	dry winding process with film stacking

**Table 2 materials-16-04842-t002:** Mean values of tensile properties of the rCF samples with different matrix systems.

Matrix (Fibre Volume Content)	Tensile Modulus (MPa)	Strain at Break (%)	Strength (MPa)
Bio-rCF (37%)	78,892.42	1.05	861.53
EP-rCF (30%)	50,271.47	1.05	551.31
PA6-rCF (30%)	45,849.74	1.04	488.10

**Table 3 materials-16-04842-t003:** Mean values of bending properties of the rCF samples with different matrix systems.

Material (Fibre Volume Content)	Bending Modulus (MPa)	Bending Strain (%)	Maximum Stress (MPa)
EP-rCF (30%)	48,212.77	1.53	643.92
PA6-rCF (30%)	41,266.07	1.56	533.54

**Table 4 materials-16-04842-t004:** Mean values of compression properties of the rCF samples with different matrix systems.

Material (Fibre Volume Content)	Compression Modulus (MPa)	Strain at Break (%)	Stress (MPa)
EP-rCF (30%)	41,197.95	1.01	375.94
PA6-rCF (30%)	37,303.34	0.91	325.87

**Table 5 materials-16-04842-t005:** Comparison of Young’s modulus of composites with rCF and HT fibres with standardised fibre volume content of 30%.

	Young’s Modulus (MPa) with rCF	Young’s Modulus (MPa) with HT Fibre	Difference (%)
EP (Thermoset)	50,271.47	54,377.52	7.55
PA6 (Thermoplastic)	45,849.74	53,161.84	13.75

## Data Availability

The data that support the findings of this study are available from the corresponding author upon reasonable request.

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
