# Peer review of "First Conclusions on Damage Behaviour of Recycled Carbon Staple Fibre Yarn Using X-ray and Acoustic Emission Techniques"

_materials, 2023, doi:10.3390/ma16134842_

Round 1

Reviewer 1 Report

The abstract is too extended. Decreasing the explanation about the background of recycling and damage behavior study is better. I think around 200 words are suitable for the abstract here.

The English writing of this paper is close to a technical report, not an academic article. The authors should pay attention to improving their English writing and overall paper structure.

The material preparation process and following results are mixed up in the experimental section. I recommend that the authors separate those parts. For example, Fiber tensile test results, FTIR, and SEM results might be moved to the result section.

I suggest the resin system selection and abbreviated sample name would be shown in a table.

Figure 7 is better in the stress-strain field.

Figure 15, the AE signal should be expressed in decibel scale.

Reviewer 2 Report

The authors used recycled CFRP for different approcahs. The paper is generally good but it needs improvement. Followings should be carried out before acceptance:

The abstract should contain important results of the study.

Language should be significantly improved. Avoid using "you" such as line 144

Most of the references are national since the publication is going to publish international journal. The authors should extend their reference list.

Line 38. CFRPs are also significantly utilzed in civil engineering applications. Following references can be added for this purposes: Optimum amount of CFRP for strengthening shear deficient reinforced concrete beams; Behavior of CFRP-strengthened RC beams with circular web openings in shear zones: Numerical study

Novelty statement is not clear. Please improve it

Add some summary for conclucision.

Quality of Fig 2 should be improved and discuessed in detail.

Add recent studies on this subject to introduction. There are many studies on the introduction for this topic.

Conclusion should be improved. The recommendation consdiering all test should be given for engineers.

Language should be significantly improved. Avoid using "you" such as line 144

Reviewer 3 Report

1.       In general, the researches related to the recycling of materials are very valuable. Here is a question. Can we know how much carbon dioxide and waste materials are produced by the methods of separating carbon fibers from the matrix? And what is the cost? Can the materials used to separate the fibers from the matrix be recycled? It is valuable if these things are explained in three to four lines.

2.       In Figure 2, it is better to remove the peaks related to probe 2 after 100 mm. The quality of the diagram is low. Improve its quality if possible.

3.       In Figure 3A, write the German labels and axes titles in English. If possible, make Figure 3A sharper.

4.       In the caption of Figure 4, replace the "top" with "a" and the "bottom" with "b".

5.       Page 8 line 188: Change scanning electron microscope to "SEM".

6.       Figure 5 is given after Figure 6 in the text. Please correct it.

7.       How are the samples shown in Figure 5 produced?

8.       Label the Figures with a, b, and...

9.       Page 15: Lines 369 and 372: Correct the cross references.

10.   In Figure 12, it seems that the scale of the zoomed parts (0.1 mm) is not correct.

11.   It is better to do a general revision about the unification of charts. Some grammatical corrections are needed.

Some grammatical corrections are needed.

Round 2

Reviewer 1 Report

My comments are well reflected and I think it is acceptable.

My comments are well reflected and I think it is acceptable.

Reviewer 3 Report

Dear Editor and Authors

The answers to the questions were convincing. All the points mentioned by me were corrected and resolved.

Regards.